# GLOBAL AND FINE-GRAINED FRAMEWORK FOR CLIP WITH CROSS-MODAL MAMBA IN FEW-SHOT IMAGE CLASSIFICATION

## ABSTRACT

CLIP is a highly efficient cross-modal text-image embedding model with remarkable generalization ability. However, the encoders in CLIP usually operate independently without dynamic cross-modal interaction, leading to suboptimal performance in few-shot classification. Therefore, we propose a Global and Fine-Grained Framework for CLIP with Cross-Modal Mamba in Few-Shot Image Classification (GF4FC). Specifically, the CLIP with Cross-Modal Mamba module (CLIMA) is conducted to leverage Transformer and Vision-Transformer to interdependently encode text and image. These cross-modal representations then serve as mutual prompts to refine the embedding space, while the proposed Cross-Modal Mamba module ensures efficient time complexity. Moreover, we design a Fine-Grained Capture module (FGC) to enhance CLIMA's image representations using a Vssm module to extract prior fine-grained information. Furthermore, the Local Feature Supplementation (LFS) module is conducted to supplement CLIP's logits with FGC-derived fine-grained representations through a residual structure. Finally, the Adaptive Logits Fusion module is constructed to dynamically fuses logits using learned adaptive weights. Experiments on seven datasets demonstrate that GF4FC achieves superior performance compared with state-of-the-art methods in few-show image classification.

## 1 INTRODUCTION

Few-Shot Learning (FSL) is a crucial approach in machine learning to tackle the problem of insufficient labeled data (Song et al., 2022), since it makes full use of existing samples to effectively obtain reliable representation and has been applied for diverse fields such as image classification (Sun et al., 2025), object detection (Köhler et al., 2024), semantic segmentation (Catalano & Matteucci, 2024) and instance segmentation (Ganea et al., 2021). Models for Vision Language Pretraining (VLP), notably CLIP (Radford et al., 2021), have significantly advanced Few-Shot Learning (FSL) by leveraging powerful representation learning capabilities derived from large-scale pretraining (Zhang et al., 2024). [1]

FSL models based on CLIP methods can be divided into three categories. (1) Prompt Learning. CLIP's unified modeling approach grounds its prompt learning techniques in text-understanding models, where "prompts" denote the textual tokens to be optimized, with the objective of refining the model's semantic representations. For instance, CoOp(Kaiyang Zhou, 2022) and CoCoOp(Zhou et al., 2022) optimize text prompts by treating them as learnable vectors to generate classification weights. However,the classification boundaries cannot be optimized, leading to confusion in the model's understanding. (2) Adapter Design. To circumvent catastrophic forgetting during data-scarce fine-tuning of CLIP, which learns unified text-image embeddings from massive pairs, an adapter-based solution freezes the pre-trained backbone and introduces lightweight adapters for new task learning (Gao et al., 2025). For example, Tip-Adapter (Zhang et al., 2021) freezes the backbone and uses lightweight modules for adaptation. Although it really improves the effectiveness of models,

---

[1]We acknowledge the use of a large language model (LLM) for assistance in polishing and refining the manuscript. However, the intellectual contribution and core significance of the work originate entirely from the authors.

it underutilizes cross-modal interactions between text and images. (3) Feature Enhancement. The internal feature representation within CLIP entails a transformation from specific, low-level details to generalized, high-level concepts, a process that dissipates fine-grained information. Concurrently, the feature emphasis is shaped by the systematic biases of the associated textual data. For example, given the text "a dog eating", the designed model usually focus on "dog" and "eating" while ignoring the color and bread of a dog, leading to degraded performance on specific tasks. Through operations such as feature decoupling, reconstruction, and reuse, enhancement techniques seek to facilitate the identification of more discriminative and task-relevant features (Bär et al., 2024) (Ye et al., 2025). For example, LDC (Li et al., 2025) decouples features to reduce category confusion. It takes the fine-grained information and one-modal information into consideration, but neglects the global feature enhancement and two-modal dynamic interaction.

To address the above issues, we proposes a Global and Fine-grained Framework based on Cross-Modal Mamba (CMM) for Few-shot Image Classification (GF4FC). This framework aims to overcome the deficiencies of existing models in establishing fine-grained semantic associations and category discrimination through deep cross-modal interaction and a global-local feature complementation mechanism. Specifically, the Cross-Modal Mamba module (CLIMA) breaks modal isolation through a bidirectional mutual prompting mechanism, enabling the collaborative operation of text and image encoders. Thereinto, text features dynamically guide image encoding, while visual features real-time correct text embeddings, achieving linear complexity interaction using State Space Model (SSM). Besides, the Fine-Grained Capture module (FGC) utilizes a Vssm structure to extract local prior visual features and captures multi-scale details through a method of multi-directional flattening and reconstructed. Moreover, the Local Feature Supplement (LFS) mechanism constructs a residual path, fusing fine-grained features extracted by FGC with global logits from CLIMA to supplement local information lost in cross-modal alignment. Furthermore, the adaptive logits fusion module dynamically balances the contributions of global and local features, automatically adjusting the decision weights of the two types of features through learnable weight factors to optimize the final classification boundary. The experimental results across seven datasets show that GF4FC has outstanding performance in few-shot image classification, outperforming state-of-the-art methods.

In summary, our key contributions are summarized as follows: (1) We introduce GF4FC, a CLIP-based method designed to enhance few-shot learning performance by integrating global feature enhancement with local feature supplementation. (2) GF4FC utilizes the State Space Model-based Mamba and Vssm methods for their high temporal efficiency, enabling efficient dual-branch feature extraction from CLIP-encoded text and imagery. (3) GF4FC consistently boosts the few-shot learning capability of CLIP across seven benchmark datasets, and it outperforms four established CLIP-based FSL baselines in parallel tests, an advantage that is particularly evident with larger sample sizes.

## 2 Approach

The details of GF4FC are introduced as follows, which also can be found in Figure 1. The proposed architecture takes as input a batch of unlabeled images list $\mathbf{x}$ of length $b$, in which the images have been cut to $224 \times 224$ size in preprocess and a set of class descriptions list $\tau$ of length $c$, in which the texts have been cut or filled to the length of 77. It outputs a probability matrix $\mathbf{S} \in \mathbb{R}^{b \times c}$, where each row corresponds to an image and each column represents the probability of that image belonging to a specific class.

The GF4FC includes CLIP with Cross-Modal Mamba (CLIMA). The CLIMA architecture comprises a text encoder and an image encoder, which embed text and images into a unified space, and comprises our novel Cross-Modal Mamba (CMM) to optimize this space by treating matched text-image pairs as mutual descriptions. This brings related embeddings closer and pushes unrelated ones farther apart. GF4FC features a Fine-Grained Capture (FGC) module. This module makes use of multilayer image features from CLIMA's image encoder. It then uses the Vssm (Liu et al., 2024) to enhance these features, and weighted sum to fuse these features. These enhanced features are used to generate logits and adaptive weight $\alpha$.

Drawing inspiration from the LDC (Li et al., 2025) architecture, GF4FC includes a Local Feature Supplementation (LFS) component, which employs a residual structure to learn fine-grained and local information from the enhanced image features, and an Adaptive Logits Fusion (ALF) module

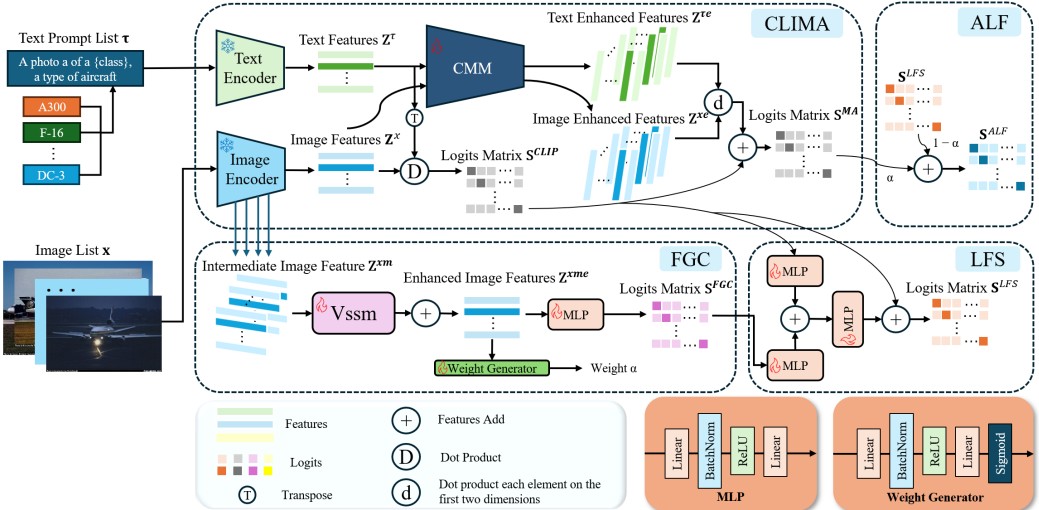

Figure 1: The architecture of our GF4FC contains four parts. (1) CLIP with Cross-Modal Mamba (CLIMA) is designed to refine embedding space, (2) Fine-Grained Capture (FGC) is aimed at extract fine-grained features from Image Encoder, (3) Local Feature Supplementation (LFS) is used to supplement CLIP's logits with local features and (4) Adaptive Logits Fusion (ALF) dynamically fuses the outputs form CLIMA and LFS.

fuses logits from enhanced features and fine-grained information using adaptively generated weights $\alpha$. Notations and symbols used are defined in Appendix.

## 2.1 CLIP WITH CROSS-MODAL MAMBA (CLIMA)

Prevailing approaches are often hampered by a static processing mechanism that treats textual and visual modalities in relative isolation. Without a dedicated mechanism for dynamic cross-modal interaction, these methods struggle to bridge the semantic gap effectively, culminating in insufficient representation learning and ambiguous model predictions (Zhang et al., 2021) (Li et al., 2025). Therefore, we propose CLIP with Cross-Modal Mamba (CLIMA), a novel framework for cross-modal modeling based on the emerging Mamba architecture, whose overall structure is depicted in Figure 1. The CLIMA framework comprises three core components: a Text Encoder, an Image Encoder, and a Cross-Modal Mamba (CMM) module.

### 2.1.1 TEXT ENCODER AND IMAGE ENCODER

The Text Encoder and Image Encoder utilize CLIP's standard Transformer and Vision Transformer architectures, based on Multi-head self-attention mechanism to generate initial text and image embeddings in a shared space. They take in Text Prompt List $\tau$ and Image List $\mathbf{x}$, and give out Text Features $\mathbf{Z}^\tau \in \mathbb{R}^{c \times e}$ and Image Features $\mathbf{Z}^x \in \mathbb{R}^{b \times e}$.

### 2.1.2 CROSS-MODAL MAMBA (CMM)

The CMM, shown in figure 2, concatenates these textual and visual features in two different orders and apply the Mamba model for sequence modeling. It takes in $\mathbf{Z}^\tau$ and $\mathbf{Z}^x$, and repeats them through Eq.1 and Eq.2.

$$\mathbf{Z}^{\tau r} \in \mathbb{R}^{b \times c \times e}, \mathbf{Z}^{\tau r}[i, :, :] = \mathbf{Z}^\tau, \quad \text{for } i = [0, 1, \ldots, b-1] \tag{1}$$

$$\mathbf{Z}^{xr} \in \mathbb{R}^{b \times c \times e}, \mathbf{Z}^{xr}[:, j, :] = \mathbf{Z}^x, \quad \text{for } j = [0, 1, \cdots, c-1] \tag{2}$$

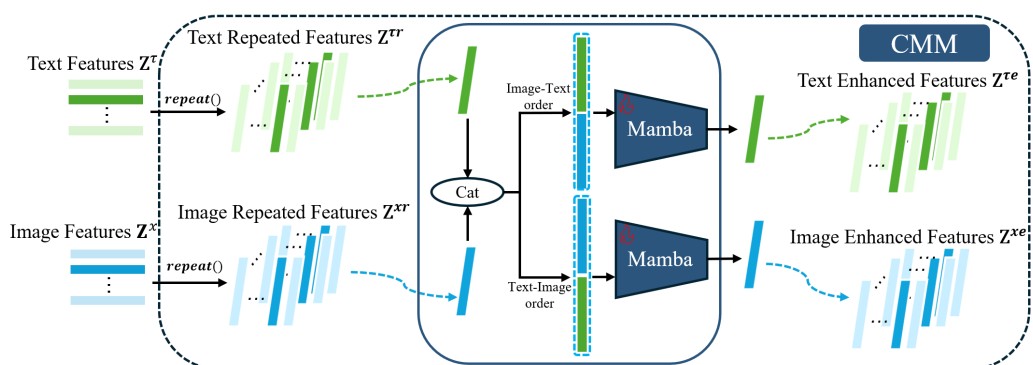

Figure 2: The Structure of CMM

$\mathbf{Z}^{\tau r}$ and $\mathbf{Z}^{xr}$ are concatenated in two ways. Then CMM conducts a bidirectional mamba fusion and gives out Text Enhanced Features $\mathbf{Z}^{\tau e} \in \mathbb{R}^{b \times c \times e}$, Image Enhanced Features $\mathbf{Z}^{xe} \in \mathbb{R}^{b \times c \times e}$ , as shown in Eq.3 and Eq.4.

$$\mathbf{Z}^{\tau e} = \mathrm{mamba}(\mathrm{cat}((\mathbf{Z}^{\tau r}, \mathbf{Z}^{xr}), dim = 1))[:, :, -1] \tag{3}$$

$$\mathbf{Z}^{xe} = \mathrm{mamba}(\mathrm{cat}((\mathbf{Z}^{xr}, \mathbf{Z}^{\tau r}), dim = 1))[:, :, -1] \tag{4}$$

where $\mathrm{cat}(\mathbf{A}, \mathbf{B}, dim = d)$ function means concatenate two matrices $\mathbf{A}$ and $\mathbf{B}$ in dimension $d$.

Mamba was originally designed for temporal modeling of sequences of embedding vectors. We leverage Mamba's sequential modeling capability to connect text and image sequences for bidirectional image-text and text-image joint temporal modeling. This process enhances text features guided by images and image features guided by text. In the CMM, each text is matched with and prompts each image and vice versa. For matching pairs, the CMM strengthens the prompted local features, while non-matching pairs cause confusion and weaken feature quality. Subsequently, during cosine similarity calculation, non-matching pairs yield lower similarity scores. Through this mechanism, we achieve enhanced positive samples, weakened negative samples, reinforced matching results, and optimized embedding space. The CMM enables text and image to prompt each other, allowing the mamba to selectively distinguish original embeddings and thereby acquire effective knowledge, bringing semantically similar text and image representations closer in the embedding space while pushing dissimilar ones further apart. The pseudocode of CMM can be found in Appendix.

After the CMM processing, the enbedding vectors for text in $\mathbf{Z}^{\tau e}$ and the enbedding vectors for images in $\mathbf{Z}^{xe}$ are dot-producted to obtain similarity indices $\Delta\mathbf{S}$ for each text-image pair:

$$\mathbf{S}^{CLIP} = \gamma \cdot \mathbf{Z}^x \cdot (\mathbf{Z}^\tau)^T \tag{5}$$

where $\gamma$ is a shared learnable parameter, aimed at scaling logits.

These indices are then added to the similarity matrix $\mathbf{S}^{CLIP}$ that wasn't processed by the CMM:

$$\Delta\mathbf{S}[i, j] = \gamma \cdot \mathbf{Z}^{xe}[i, j, k] \cdot \mathbf{Z}^{\tau e}[i, j, k] \quad \text{for } i = 0, 1, \ldots, b - 1 \text{ and } j = 0, 1, \cdots, c - 1 \tag{6}$$

$$\mathbf{S}^{MA} = \Delta\mathbf{S} + \mathbf{S}^{CLIP} \tag{7}$$

The resultant Logits Matrix $\mathbf{S}^{MA}$ is fine-tuned based on $\mathbf{S}^{CLIP}$ with global features.

## 2.2 Fine-Grained Capture (FGC)

The FGC module is designed to capture fine-grained and local information to supplement the classification. Specifically, the FGC module first takes advantages of Intermediate Image Features

$\mathbf{Z}^{xm} \in \mathbb{R}^{l \times b \times h \times w}$ (where we set $l = 4$) from the Image Encoder. It inputs it into the Vssm (Liu et al., 2024), where feature maps pass through an SS2D state space model to capture long-range dependencies and enhance global contextual information:

$$\text{for each } \mathbf{Z} \in \mathbf{Z}^{xm}, \quad \mathbf{Z}^e = \text{Append}(\mathbf{Z}^e, Vssm(\mathbf{Z})) \tag{8}$$

where $\mathbf{Z}^e$ is an empty list and $\text{Append}(\mathbf{A}, \mathbf{B})$ is a function that appends element $\mathbf{B}$ to list $\mathbf{A}$.

Afterward, the feature maps are fused by weighted sum with weight $\beta_1, \beta_2, \beta_3, \beta_4$:

$$\mathbf{Z}^{xme} = \beta_1 \mathbf{Z}^e[0] + \beta_2 \mathbf{Z}^e[1] + \beta_3 \mathbf{Z}^e[2] + \beta_4 \mathbf{Z}^e[3] \tag{9}$$

and then projected into the embedding space to generate the Enhanced Image Features $\mathbf{Z}^{xme} \in \mathbb{R}^{b \times e}$. Subsequently, $\mathbf{Z}^{xme}$ goes through a MLP to produce Logits Matrix $\mathbf{S}^{FGC}$:

$$\mathbf{S}^{FGC} = MLP(\mathbf{Z}^{xme}) \tag{10}$$

Through this process, FGC extracts fine-grained features from the prior features obtained before CMM. This enhances the utilization of feature information and serves as a foundation for the subsequent LFS module, ensuring more effective feature refinement and utilization. The pseudocode of FGC can be found in Appendix.

## 2.3 Local Feature Supplementation (LFS)

After obtaining $\mathbf{S}^{FGC}$ from the FGC module, LFS uses it to perform feature supplementation through three MLPs and a residual structure. Specifically, $\mathbf{S}^{FGC}$ first undergoes dimensionality expansion via an MLP to optimize the spatial and manifold structure of fine-grained features in $\mathbf{S}^{FGC}$, enhancing their usability. Simultaneously, $\mathbf{S}^{CLIP}$ passes through another MLP to project it onto the same dimension as the expanded $\mathbf{S}^{FGC}$. The two outputs are then added together and fed into a third MLP to jointly learn the missing fine-grained features in $\mathbf{S}^{CLIP}$. Finally, a residual connection with $\mathbf{S}^{CLIP}$ is applied to supplement local features:

$$\mathbf{S}^{LFS} = \mathbf{S}^{CLIP} + MLP^3(MLP^1(\mathbf{S}^{FGC}) + MLP^2(\mathbf{S}^{CLIP})) \tag{11}$$

This module uses $\mathbf{S}^{CLIP}$ as a dynamic prompt to supplement the fine-grained and local features in $\mathbf{S}^{FGC}$. Consequently, the LFS module learns to recognize feature-supplementation patterns. This strengthens the utilization of CLIP-encoded image features, boosts the model's ability to adapt to new data, and enhances few-shot performance.

## 2.4 Adaptive Logits Fusion (ALF)

Finally, the ALF module adaptively fuses $\mathbf{S}^{MA}$ and $\mathbf{S}^{LFS}$ to obtain the final adaptive Logits Matrix $\mathbf{S}^{ALF}$. Specifically, we use a Weight Generator to produce an adaptive weight $\alpha$ from enhanced features $\mathbf{Z}^{xme}$, then apply $\alpha$ for a weighted fusion of $\mathbf{S}^{FGC}$ and $\mathbf{S}^{LFS}$:

$$\mathbf{S}^{ALF} = \alpha \mathbf{S}^{MA} + (1 - \alpha)\mathbf{S}^{LFS}, \alpha = WeightGenerator(\mathbf{Z}^{xme}) \tag{12}$$

Via the above ALF module, our approach can adaptively supplement missing fine-grained and local features. This makes the final $\mathbf{S}^{ALF}$ more robust and accurate. The core lies in using $\alpha$ to achieve more reasonable fusion.

## 2.5 Loss Function

In GF4FC, we train CMM of CLIMA, Vssm, MLP, Weight Generator in FGC and MLPs in LFS. Specifically, our goal is to make the Logits Matrix $\mathbf{S}^{MA}$ output by CLIMA achieve good classification accuracy and realize global feature supplementation to ensure accurate classification through

---

**Algorithm 1** The procedure of the proposed GF4FC.

---

**Require:** Text prompt list $\tau$, image List $\mathbf{x}$, image label list $\mathbf{y}$, trade-off parameter $\lambda$, weight $\beta_1, \beta_2, \beta_3, \beta_4$, a learnable logit scale $\gamma$

**Ensure:** Adaptive Logits $\mathbf{S}^{ALF}$.

1: $\mathbf{Y} \leftarrow$ using $\mathbf{y}$ by Eq.(16);
2: $\mathbf{Z}^\tau \leftarrow TextEncoder(\tau)$;
3: $\mathbf{Z}^x$ and $\mathbf{Z}^{xm} \leftarrow ImageEncoder(\mathbf{x})$;
4: $\mathbf{S}^{CLIP} \leftarrow \gamma \cdot \mathbf{Z}^x \cdot (\mathbf{Z}^\tau)^T$;
5: $\mathbf{Z}^{\tau e}$ and $\mathbf{Z}^{xe} \leftarrow$ using $\mathbf{Z}^\tau$ and $\mathbf{Z}^x$ by Alg.2;
6: $\Delta \mathbf{S} \leftarrow \gamma \cdot einsum('ijk, ijk- > ij', \mathbf{Z}^{xe}, \mathbf{Z}^{\tau e})$;
7: $\mathbf{S}^{MA} \leftarrow \Delta \mathbf{S} + \mathbf{S}^{CLIP}$;
8: $\mathbf{S}^{FGC}, \mathbf{Z}^{xme} \leftarrow$ using $\mathbf{Z}^{xm}$ and weight $\beta_1, \beta_2, \beta_3, \beta_4$ by Alg.3
9: $\mathbf{S}^{LFS} \leftarrow$ using $\mathbf{S}^{FGC}$ and $\mathbf{S}^{CLIP}$ by Eq.(11);
10: $\mathbf{S}^{ALF} \leftarrow$ using $\mathbf{Z}^{xme}, \mathbf{S}^{MA}$ and $\mathbf{S}^{LFS}$ by Eq.(12);
11: $\mathcal{L}_{CE} \leftarrow$ using $\mathbf{S}^{ALF}, \mathbf{S}^{LFS}, \mathbf{S}^{FGC}$ and $\mathbf{Y}$ by Eq.(13);
12: $\mathcal{L}_{Sim} \leftarrow$ using $\mathbf{S}^{LFS}, \mathbf{S}^{FGC}$ and $\mathbf{S}^{CLIP}$ by Eq.(14);
13: $\mathcal{L} \leftarrow$ using $\mathcal{L}_{CE}, \mathcal{L}_{Sim}$ and $\lambda$ by Eq.(15);
14: $\mathcal{L}.backward()$;
15: Update all learnable parameters in GF4FC by AdamW;
16: **return** $\mathbf{S}^{ALF}$.

---

$\mathbf{S}^{FGC}$, $\mathbf{S}^{LFS}$, and $\mathbf{S}^{ALF}$. So we employ Cross-Entropy Loss to assess the discrepancy between $\mathbf{S}^{MA}$, $\mathbf{S}^{FGC}$, $\mathbf{S}^{LFS}$, and $\mathbf{S}^{ALF}$:

$$\mathcal{L}_{\text{totalCE}} = \mathcal{L}_{\text{CEMA}}(\mathbf{S}^{MA}, \mathbf{Y}) + \mathcal{L}_{\text{CEFGC}}(\mathbf{S}^{FGC}, \mathbf{Y}) + \mathcal{L}_{\text{CELFS}}(\mathbf{S}^{LFS}, \mathbf{Y}) + \mathcal{L}_{\text{CEALF}}(\mathbf{S}^{ALF}, \mathbf{Y}) \quad (13)$$

where $\mathbf{Y}$ is the Sample Probability Distribution Matrix, and the detailed definition can be found in Appendix.

Meanwhile, to prevent over-enhancement of features and overfitting to samples, we use the L1 Loss to calculate the difference between $\mathbf{S}^{FGC}$ and $\mathbf{S}^{CLIP}$, and between $\mathbf{S}^{LFS}$ and $\mathbf{S}^{CLIP}$:

$$\mathcal{L}_{\text{Sim}} = \mathcal{L}_{\text{L1FGC}}(\mathbf{S}^{FGC}, \mathbf{S}^{CLIP}) + \mathcal{L}_{\text{L1LFS}}(\mathbf{S}^{LFS}, \mathbf{S}^{CLIP}) \quad (14)$$

For ultimate loss, we introduce a trade-off parameter to balance the impact between $\mathcal{L}_{\text{totalCE}}$ and $\mathcal{L}_{\text{Sim}}$:

$$\mathcal{L}_{\text{final}} = \mathcal{L}_{\text{totalCE}} + \lambda \mathcal{L}_{\text{Sim}} \quad (15)$$

where $\lambda$ is a trade-off parameter.

## 3 EXPERIMENTS

### 3.1 DATASETS

To verify our method's effectiveness, we employed seven classification datasets, including Caltech101 (Fei-Fei et al., 2004), DTD (Cimpoi et al., 2014), EuroSAT (Helber et al., 2019), FGVCAircraft (Maji et al., 2013), Flowers102 (Nilsback & Zisserman, 2008), Food101 (Bossard et al., 2014), and OxfordPets (Parkhi et al., 2012). Detailed imformation can be found in Appendix.

### 3.2 COMPARISON METHODS

We compared our method with several baseline methods, including SuS-X (Udandarao et al., 2023), CoOp (Zhou et al., 2022), Tip-Adapter, and Tip-Adapter-F (Zhang et al., 2021). The introduction of these methods can be found in Appendix.

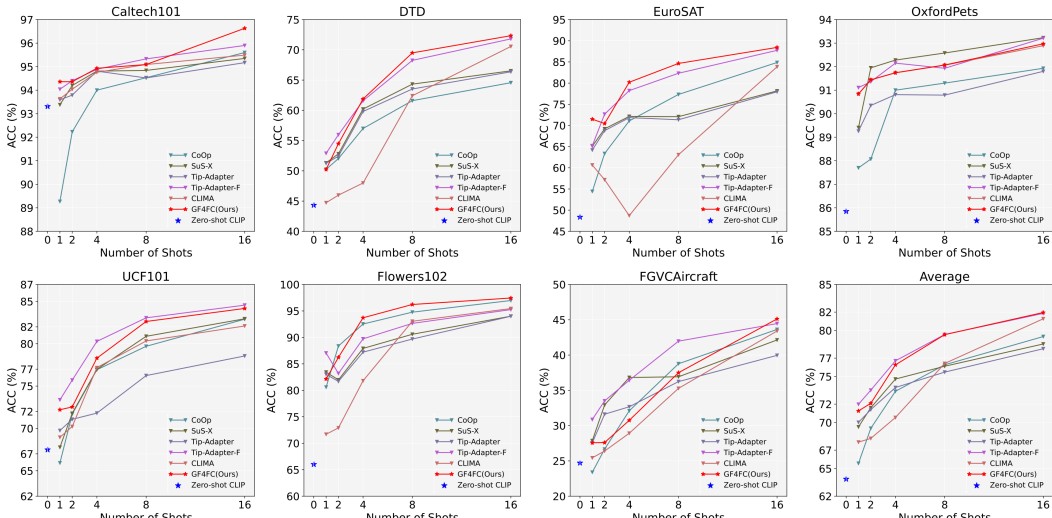

Figure 3: Classification performance of different methods on 7 datasets, and the last one is the average performance on 7 datasets.

## 3.3 SETTINGS

All experiments were conducted with an NVIDIA GeForce RTX 4090 GPU, using Ubuntu 22.04. In GF4FC, the parameters we prepared for the CLIMA module are pre-trained. Among them, the text and image encoders of CLIMA are initialized with parameters from ViT-B/16 (Radford et al., 2021), and these parameters remain frozen throughout subsequent experiments. For CMM, FGC, LFS and ALF, the parameters are initialized randomly. For fairness, other comparison methods were all backed by ViT-B/16 as backbone. We set the batch size to $64$ and the learning rate to $10^{-4}$, using cosine annealing for $50$ epochs. The optimizer was AdamW with weight decay at $10^{-4}$ and eps at $10^{-4}$, hyperparameter $\lambda$ is 1. We conducted experiments for 1-Shot, 2-Shot, 4-Shot, 8-Shot, and 16-Shot scenarios. For other comparison methods, hyperparameters followed the official recommendations. We used accuracy as the metric for comparison.

## 3.4 EXPERIMENT RESULTS AND ANALYSIS

To evaluate GF4FC's effectiveness, we tested it on seven image classification datasets and compared its classification accuracy with several SOTA CLIP-based FSL methods using ViT-B/16 as the backbone. The result is shown in Figure 3.

Firstly, we examined how our method performed across different datasets. Results showed that on Caltech101, the 16-Shot setting was optimal, with an accuracy 0.73% higher than the second-best method. On DTD, EuroSAT, and Flower102, higher-shot settings were more effective. At 8-Shot, our method outperformed the second-best by 1.24%, 2.32%, and 3.57%, respectively, and at 16-Shot, by 0.53%, 0.69%, and 2.19%. However, on OxfordPets, UCF101, and FGVCAircraft, our method underperformed compared to Tip-Adapter-F. We attribute this to the characteristics of the datasets. Caltech101, DTD, EuroSAT, and Flower102 have clear global structures and moderate fine-grained differences. In contrast, OxfordPets, UCF101, and FGVCAircraft require focus on local features or extremely fine-grained details. GF4FC, which emphasizes the complementary use of global and local features and cross-modal dynamic interaction, suits tasks needing semantic alignment and context modeling. But in extremely fine-grained or local feature - dominant tasks, cross-modal interaction may add noise or fail to capture key local areas.

Secondly, we looked into our model's average performance across all datasets. The average classification accuracy, as shown in Table 1, indicates better performance at higher shots, with 16-Shot being the best, yet slightly lower than Tip-Adapter-F at lower shots. We think this is due to GF4FC's multiple trainable modules (CLIMA, Vssm and MLP in FGC, MLP in LFS, and the weight generator in ALF), which have many parameters and need enough data to learn meaningful feature transfor-

mation and interaction. With only 1-2 samples per class, the complex model may overfit to the noise or unique features of these few samples, reducing generalization. As the number of samples per class increases to 8 or 16, these modules can learn meaningful patterns, and the advantages of the large model capacity emerge. Ample samples allow CLIMA's cross-modal interaction to better learn robust and discriminative text-image correlation rules.

GF4FC's key innovation lies in its dual-path design of global feature enhancement (CLIMA) and local feature supplementation (FGC+LFS). When data is sufficient, this design can learn richer and more comprehensive representations than a single feature stream. CLIMA brings semantically similar text and image representations closer through cross-modal interaction, optimizing the global embedding space. FGC and LFS supplement local details that CLIP might miss, crucial for distinguishing similar categories. ALF dynamically fuses global and local information based on input images. The more data, the better the weight generator in ALF can be trained, leading to more accurate decisions. GF4FC is designed for practical few-shot learning scenarios with about ten samples per class, where its advantages can be fully utilized.

Table 1: The average results of all methods over 7 datasets.

| Methods | K-Shot Accuracy (%) | | | | | |
|---------|------|------|------|------|------|------|
|         | 0    | 1    | 2    | 4    | 8    | 16   |
| ZS-CLIP | 63.86 | - | - | - | - | - |
| CoOp | - | 65.57 | 69.38 | 73.40 | 76.28 | 79.34 |
| Tip-Adapter | - | 70.02 | 71.41 | 73.81 | 75.48 | 78.02 |
| Tip-Adapter-F | - | 71.99 | 73.52 | 76.71 | 79.57 | 81.86 |
| SuS-X | - | 69.53 | 71.61 | 74.73 | 76.12 | 78.54 |
| CLIMA | - | 62.87 | 65.29 | 67.64 | 69.37 | 71.36 |
| GF4FC (Our) | - | **71.26** | **72.09** | **76.27** | **79.55** | **81.95** |

## 3.5 ABLATION EXPERIMENTS

To analyze the effect of our modules and settings, we conduct a series of ablation experiments over seven datasets.

### 3.5.1 EFFECT ANALYSIS OF EACH MODULE

In our work, we propose four key modules: CLIMA, FGC, LFS, and ALF. To investigate their individual contributions, we conducted a series of ablation experiments. The results are presented in Table 2. As shown in it, GF4FC with ViT-B/16 backbone achieves a significant improvement, raising the average accuracy by 19.39%. Each module contributes effectively. Adding the CLIMA module boosts average accuracy by 14.82%, while its removal causes a 2.72% drop. FGC and LFS together increase accuracy by 16.67%, and removing them leads to a 5.11% decrease. The ALF module further enhances accuracy when combined with others. These results highlight the model's efficiency in improving global and fine-grained feature processing.

Table 2: The average results of each module over 7 datasets.

| Module | | | | | Backbone |
|--------|-------|-----|-----|-----|----------|
| CLIP | CLIMA | FGC | LFS | ALF | ViT-B/16 |
| ✓ | | | | | 62.56 |
| ✓ | ✓ | | | | 77.38 |
| ✓ | | ✓ | | | 79.53 |
| ✓ | | ✓ | ✓ | | 79.23 |
| ✓ | ✓ | ✓ | ✓ | ✓ | 81.95 |

### 3.5.2 ABLATION STUDY OF CLIMA

In the ablation study of the CLIMA module, we explored two approaches: using the cosine similarity result $\Delta\mathbf{S}$ of enhanced features $\mathbf{Z}^{\tau e}$ and $\mathbf{Z}^{xe}$ directly as $\mathbf{S}^{MA}$, or combining it with $\mathbf{S}^{CLIP}$ to derive $\mathbf{S}^{MA}$, as shown in Table 3. The results indicate that integrating $\Delta\mathbf{S}$ as a global feature supplement for fine-tuning is more effective. This is likely because $\Delta\mathbf{S}$ captures unique semantic information that enhances the discriminative power of the model. By combining it with $\mathbf{S}^{CLIP}$, we enrich the feature space, allowing the model to leverage both the enhanced semantic information from $\Delta\mathbf{S}$ and the robust, pre-trained features from CLIP. This synergy leads to better performance across both CLIMA and GF4FC datasets. Moreover, the efficacy of using CMM for global feature enhancement is underscored by these results. CMM appears to effectively refine the global features, making them more suitable for the specific fine-tuning tasks at hand. The improvement in accuracy when using the combined approach suggests that CMM helps in creating a more balanced and informative feature representation, which is crucial for achieving higher classification accuracy. This approach not only boosts the model's ability to generalize but also highlights the importance of strategically integrating different feature sources to maximize performance.

Table 3: Ablation study of CLIMA module.

| $\mathbf{S}^{MA}$ | CLIMA | GF4FC |
|:---:|:---:|:---:|
| $\Delta\mathbf{S}$ | 70.73 | 80.46 |
| $\Delta\mathbf{S} + \mathbf{S}^{CLIP}$ | **77.38** | **81.95** |

### 3.5.3 ABLATION STUDY OF FGC

Table 4: The average results of parameters study.

| $\beta_1$ | $\beta_2$ | $\beta_3$ | $\beta_4$ | ACC |
|:---:|:---:|:---:|:---:|:---:|
| 0.4 | 0.3 | 0.2 | 0.1 | 80.90 |
| 0.25 | 0.25 | 0.25 | 0.25 | 80.92 |
| 0.1 | 0.2 | 0.3 | 0.4 | **80.97** |

In the ablation study of FGC module, we explored different fine-grained feature extraction method. We have tried to extract multi-layer features from CLIP's Image Encoder, and define balance parameters $\beta_1$, $\beta_2$, $\beta_3$ and $\beta_4$ to control the contribution of the feature from different layer. We test different combination of features from the last four layers of CLIP's Image Encoder, and the results are shown below in Table 4. As shown in it, GF4FC focuses mainly on the intermediate feature of CLIP's deep layer for fine-grained and local information capture, achieving the highest accuracy of 80.97%. This strategy is effective for several reasons. Firstly, the deep layer's features capture high-level semantics crucial for classification. Secondly, these features are task-specific and refined, making them ideal for fine-grained distinctions. Thirdly, using deeper layer reduces model complexity and avoids noise from shallower layers.

## CONCLUSION

In this paper, we propose a novel method called Global and Fine-Grained Framework for CLIP with Cross-modal Mamba in Few-Shot Learning (GF4FC) to better solve Few-Shot Classification problems. GF4FC uses Cross-Modal Mamba (CMM) for global feature enhancement, and Fine-Grained Capture (FGC) and Local Feature Supplementation (LFS) modules for fine-grained and local feature supplementation. These two paths are dynamically fused via the ALF module, boosting CLIP's few-shot classification performance. Experiments on seven datasets, compared with four methods, validate our framework's effectiveness.

In the future, we plan to optimize the mutual prompting mechanism in the CLIMA module. And extend GF4FC to dense prediction tasks such as object detection (*e.g.*COCO) and segmentation (*e.g.*ADE20K) where fine-grained alignment is critical.

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

# A  APPENDIX

# B  RELATED WORK

In Related Work, we are going to introduce the three main aspects of CLIP-based FSL methods: Prompt Learning, Adapter Design and Feature Enhancement.

## B.1  PROMPT LEARNING IN CLIP-BASED FSL

Text prompts, sentence-like instructions given to the language branch of VLP models, help them understand tasks. Prompts can be manually designed for downstream tasks or automatically learned during fine-tuning, known as Prompt Learning. Many works adapt VLP models by learning prompts in end-to-end training. CoOpKaiyang Zhou (2022) fine-tunes CLIP's language branch by optimizing continuous prompt vectors for few-shot transfer, while CoCoOpZhou et al. (2022) conditions prompts on image instances to address CoOp's generalization issues. MaPLekhattak et al. (2023) enhances consistency between visual and language representations through multimodal prompt learning across both branches. However, these methods, not optimizing classification boundaries during pre-training, can cause category confusion in downstream tasks.

## B.2  ADAPTER DESIGN IN CLIP-BASED FSL

In the field of natural language processing (NLP), fine-tuning large pre-trained models is an effective way to transfer knowledge to downstream tasks. However, when facing many downstream tasks, this method lacks parameter efficiency: each task requires a brand-new model. As an alternative, (Houlsby et al., 2019) proposed using adapter modules for transfer learning. These adapter modules can produce compact and scalable models, adding only a small number of trainable parameters per task and allowing new tasks to be added without reprocessing previous ones. The original network parameters remain fixed, achieving high parameter sharing. Adapters, as an alternative to full model fine-tuning, have been widely used in many areas, including FSL. For example, Tip-Adapter(Zhang et al., 2021) uses the features and labels of a few-shot support set to build a non-parametric classifier based on caching, which is then weighted and fused with CLIP's zero-shot classifier. It requires no training and is fast. Tip-Adapter-F(Zhang et al., 2021) enhances this by adding a tiny MLP adapter module for end-to-end fine-tuning. CALIP(Guo et al., 2022) adds adapter layers on frozen CLIP features and uses zero-shot CLIP predictions as a prior to modulate adapter outputs via a prior-adapter module, reducing overfitting to few-shot data. However, these approaches still underuse the cross-modal interaction between text and image.

## B.3  FEATURE ENHANCEMENT IN CLIP-BASED FSL

Feature enhancement improves feature extraction to boost model performance. In CLIP-Based FSL, it focuses on optimizing cross-modal features to address category confusion and domain differences. Key methods include feature decoupling and reconstruction. For example, LDC(Li et al., 2025) decomposes classification logits, isolates discriminant and confusion components, and suppresses noise. It focus on multi-layer image features, but may ignore the global information enhancement in classification and may not effectively utilize the rich cross-modal potential of CLIP. Cross-modal Feature Refiner(Zhang et al., 2025) uses a lightweight Transformer to refine visual features with textual guidance. It is a dynamical refiner that can enhance the fine-grained features of images, but ignores that images can also prompt texts.

To overcome these limitations, we propose GF4FC. GF4FC introduces the Cross-Modal Mamba module to interdependently encode text and image, using cross-modal representations as mutual prompts to refine the embedding space. This approach not only enhances the model's adaptability and generalization ability but also improves classification performance by strengthening dynamic cross-modal interaction.

Table 5: The Descriptions of Used Notations.

| Notations | Descriptions |
|---|---|
| $\mathbf{X}^n$ | matrix $\mathbf{X}$ named n |
| $\mathbf{X}^n_{i,\cdot}$ | the $i$-th row of $\mathbf{X}^n$ |
| $\mathbf{X}^n_{\cdot,j}$ | the $j$-th column of $\mathbf{X}^n$ |
| $\mathbf{x}_i$ | the $i$-th element of $\mathbf{x}$ |
| $rank(\mathbf{X})$ | the rank of $\mathbf{X}$ |
| $\mathbf{X}^T$ | the transpose of $\mathbf{X}$ |
| $tr(\mathbf{X})$ | the trace of $\mathbf{X}$ |
| $\mathbf{X}^{-1}$ | the inverse of $\mathbf{X}$ |

Table 6: The Definitions of Used Symbols

| Symbols | Definitions |
|---|---|
| $c$ | the number of texts |
| $b$ | the number of images |
| $h$ | height |
| $w$ | width |
| $cls$ | the number of classes |
| $e$ | embedding dimension |
| $l$ | the number of layers |
| $s$ | the length of the sequence |
| $\mathrm{MLP}^n$ | a multilayer perceptron named $n$ |

## C  APPROACH

### C.1  NOTATIONS AND DEFINITIONS

In this paper, matrices, vectors, and scalars are defined in boldface uppercase, boldface lowercase, and normal italic, respectively, *e.g.* $\mathbf{X}$, $\mathbf{x}$, and $x$. In addition, some notations are listed in Table 5.

In addition to the above notation conventions, we use some symbols with specific meanings as shown in the Table 6.

### C.2  DATA PREPROCESSING

GF4FC is a deep learning architecture that handles both images and text, so data preprocessing is divided into text and image parts.

For text preprocessing, picture-text datasets have natural text descriptions but in classification datasets, text descriptions are generated via "label + template" way, as shown in Table 7.

Table 7: The Templates for Classification Datasets

| Datasets | Templates |
|---|---|
| Caltech101 | a photo of a {label}. |
| DTD | {label} texture. |
| EuroSAT | a centered satellite photo of {label}. |
| FGVCAircraft | a photo of a {label}, a type of aircraft. |
| Food101 | a photo of {label}, a type of food. |
| OxfordPets | a photo of a {label}, a type of pet. |
| Flower102 | a photo of a {label}, a type of flower. |

After that, text descriptions are tokenized using a BPE tokenizer(Shibata et al., 1999), which splits text into common subword units based on frequency. Token sequences longer than the $max\_len$ are

truncated to $max\_len$, and those shorter than $max\_len$ are padded with 0 to reach $max\_len$. We set $max\_len$ to 77.

For image preprocessing, images are first resized to 224×224 using bicubic interpolation, then converted to RGB format and normalized with mean (0.48145466, 0.4578275, 0.40821073) and standard deviation (0.26862954, 0.26130258, 0.27577711).

## C.3 PSEUDOCODE

The pseudocode of CMM module and FGC module are shown in Alg.2 and Alg.3 respectively.

---

**Algorithm 2** The procedure of CMM.

---

**Require:** Text Features $\mathbf{Z}^\tau$, Image Features $\mathbf{Z}^x$.
**Ensure:** Txt Enhanced Feats $\mathbf{Z}^{\tau e}$, Img Enhanced Feats $\mathbf{Z}^{xe}$.
1: $\mathbf{Z}^{\tau r} \leftarrow$ repeat_interleave$(\mathbf{Z}^\tau, \mathbf{Z}^x.shape[0], dim = 0)$;
2: $\mathbf{Z}^{xr} \leftarrow$ repeat$(\mathbf{Z}^x, \mathbf{Z}^\tau.shape[0], dim = 1)$;
3: $\mathbf{Z}^{\tau e} \leftarrow$ mamba$(cat((\mathbf{Z}^{\tau r}, \mathbf{Z}^{xr}), dim = 1))[:, :, -1]$;
4: $\mathbf{Z}^{xe} \leftarrow$ mamba$(cat((\mathbf{Z}^{xr}, \mathbf{Z}^{\tau r}), dim = 1))[:, :, -1]$;
5: **return** $\mathbf{Z}^{\tau e}, \mathbf{Z}^{xe}$.

---

**Algorithm 3** The procedure of FGC.

---

**Require:** Intermediate Image Features $\mathbf{Z}^{xm}$, weight $\beta_1, \beta_2, \beta_3, \beta_4$.
**Ensure:** Logits Matrix $\mathbf{S}^{FGC}$, Enhanced Image Features $\mathbf{Z}^{xme}$.
1: $\mathbf{Z}^e \leftarrow initList()$;
2: **for** $\mathbf{Z}$ in $\mathbf{Z}^{xm}$ **do**
3: $\quad \mathbf{Z}^e.append(Vssm(\mathbf{Z}))$;
4: **end for**
5: $\mathbf{Z}^{xme} \leftarrow \beta_1 \mathbf{Z}^e[0] + \beta_2 \mathbf{Z}^e[1] + \beta_3 \mathbf{Z}^e[2] + \beta_4 \mathbf{Z}^e[3]$;
6: $\mathbf{S}^{FGC} \leftarrow MLP(\mathbf{Z}^{xme})$;
7: **return** $\mathbf{S}^{FGC}, \mathbf{Z}^{xme}$.

---

# D EXPRIMENTS

## D.1 DATASETS

To verify our method's effectiveness, we employed seven classification datasets, including Caltech101(Fei-Fei et al., 2004), DTD(Cimpoi et al., 2014), EuroSAT(Helber et al., 2019), FGVCAircraft(Maji et al., 2013), Flowers102(Nilsback & Zisserman, 2008), Food101(Bossard et al., 2014), and OxfordPets(Parkhi et al., 2012). Following the practices of prior methods like CoOp(Kaiyang Zhou, 2022), we divided each dataset into training, validation, and test sets. The details are shown in Table 8.

Table 8: Datasets Statistics

| Dataset | Classes | Train | Val | Test |
|---|---|---|---|---|
| Caltech101 | 100 | 4,128 | 1,649 | 2,465 |
| Flowers102 | 102 | 4,093 | 1,633 | 2,463 |
| FGVCAircraft | 100 | 3,334 | 3,333 | 3,333 |
| OxfordPets | 37 | 2,944 | 736 | 3,669 |
| DTD | 47 | 2,820 | 1,128 | 1,692 |
| EuroSAT | 10 | 13,500 | 5,400 | 8,100 |
| UCF101 | 101 | 7,639 | 1,898 | 3,783 |

## D.2 SETTINGS

All experiments were conducted on an Intel Xeon Gold 6430 CPU with an NVIDIA GeForce RTX 4090 GPU, using Ubuntu 22.04. For our GF4FC, experiments were in Python 3.8.20, with PyTorch 1.13.0+cu117 as the framework. For other comparison methods, the PyTorch environment and Python version followed the official recommendations.

## D.3 COMPARISON METHODS

We compared our method with several baseline methods, including SuS-X (Udandarao et al., 2023), CoOp (Zhou et al., 2022), Tip-Adapter, and Tip-Adapter-F (Zhang et al., 2021). We used accuracy as the metric for comparison.

1. SuS-X enables a novel paradigm called "name-only transfer," meaning the only knowledge about the downstream task during fine-tuning is the name of the target category. Consisting of two key components: SuS and TIP-X, it neither requires fine-tuning with dense data nor demands expensive labeled data. In this way, a new fine-tuning paradigm is established.

2. CoOp is a method designed to adapt visual-language pre-trained models, such as CLIP, to downstream tasks. Specifically, CoOp employs learnable vectors to model the words in a prompt, with the pre-trained model's parameters kept fixed throughout the process. To address diverse image recognition tasks, the authors have provided two implementations of CoOp: unified context and class-specific context.

3. CoOp is a method designed to adapt visual-language pre-trained models, such as CLIP, to downstream tasks. Specifically, CoOp employs learnable vectors to model the words in a prompt, with the pre-trained model's parameters kept fixed throughout the process. To address diverse image recognition tasks, the authors have provided two implementations of CoOp: unified context and class-specific context.

## D.4 ABLATION EXPERIMENTS

### D.4.1 EFFECT ANALYSIS OF EACH MODULE

As shown in Table 2, our GF4FC significantly improves the average accuracy by 19.39% with ViT-B/16 as the backbone. All four modules demonstrate efficacy. After adding the CLIMA module, the average classification accuracy increases by 14.82%. Removing CLIMA from the entire architecture leads to a 2.72% accuracy drop, indicating that CLIMA enhances global feature perception and improves classification. FGC and LFS can increase accuracy by 16.67%. Removing them causes a 5.11% accuracy decrease, suggesting the FGC-LFS route effectively captures fine-grained and local features, boosting the model's discrimination. The ALF module dynamically combines CLIMA and FGC-LFS, further improving accuracy. This shows the model efficiently performs parallel enhancement and fusion of global and fine-grained features.

## D.5 ABLATION IN LFS

### D.5.1 STRATEGY 1 IN LFS

Shown in figure 4, in Strategy 1, we abandon complex models and use MLPs for knowledge integration. $\mathbf{S}^{FGC}$ is first expanded via an MLP to enhance its fine-grained features' structure and usability. Concurrently, $\mathbf{S}^{CLIP}$ is projected to the same dimension using another MLP. The combined outputs are then fed into a third MLP to learn missing fine-grained features in $\mathbf{S}^{CLIP}$. Finally, a residual connection with $\mathbf{S}^{CLIP}$ supplements local features.

### D.5.2 STRATEGY 2 IN LFS

Shown in figure 5, in Strategy 2, we aim to boost performance by strengthening existing fine-grained knowledge. We use a Mamba module, which takes the concatenation of $\mathbf{S}^{CLIP}$ and $\mathbf{S}^{FGC}$ as input and outputs an optimized $\mathbf{S}^{FGC}$. Our goal is to model the use of $\mathbf{S}^{CLIP}$ to guide the optimization of the geometric structure of knowledge in $\mathbf{S}^{FGC}$. This allows us to decouple the fine-grained knowledge needed by $\mathbf{S}^{CLIP}$ and improve knowledge utilization. Then, using the optimized $\mathbf{S}^{FGC}$

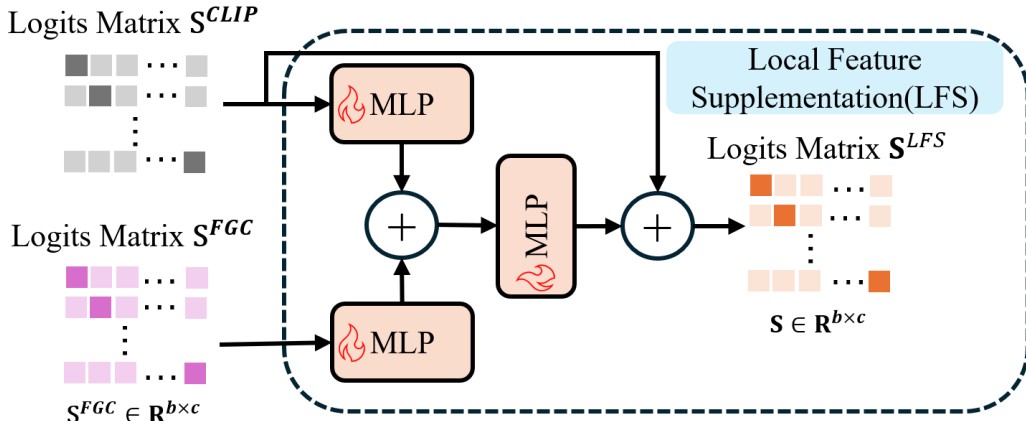

Figure 4: Strategy 1 of LFS

as fine-tuning knowledge and adding it via a residual connection to $\mathbf{S}^{CLIP}$, we obtain the final output $\mathbf{S}^{LFS}$.

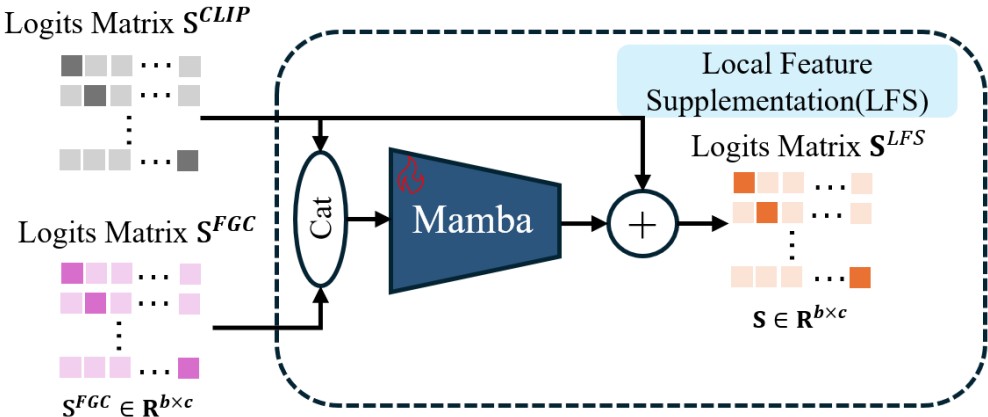

Figure 5: Strategy 2 of LFS

### D.5.3 STRATEGY 3 IN LFS

Shown in figure 6, in Strategy 3, we also aim to enhance performance by reinforcing existing fine-grained knowledge. Here, we employ a CrossAttention module, where $\mathbf{S}^{CLIP}$ acts as the query, and $\mathbf{S}^{FGC}$ serves as both the key and value, outputting an optimized $\mathbf{S}^{FGC}$. Through this cross-attention mechanism, we hope that $\mathbf{S}^{CLIP}$ can guide the targeted optimization of knowledge manifolds in $\mathbf{S}^{FGC}$. Similar to Strategy 2, we use the optimized $\mathbf{S}^{FGC}$ as fine-tuning knowledge, add it via a residual connection to $\mathbf{S}^{CLIP}$, and get the final output $\mathbf{S}^{LFS}$.

### D.5.4 STRATEGY 4 IN LFS

Shown in figure 7, Strategy 4 combines the methods of Strategies 2 and 3. First, we optimize the structure of $\mathbf{S}^{FGC}$ using the Mamba module. Then, we add a CrossAttention module, again using $\mathbf{S}^{CLIP}$ as the query and $\mathbf{S}^{FGC}$ as the key and value, to further optimize $\mathbf{S}^{FGC}$. Finally, using the optimized $\mathbf{S}^{FGC}$ as fine-tuning knowledge and combining it with $\mathbf{S}^{CLIP}$ via a residual connection, we obtain $\mathbf{S}^{LFS}$.

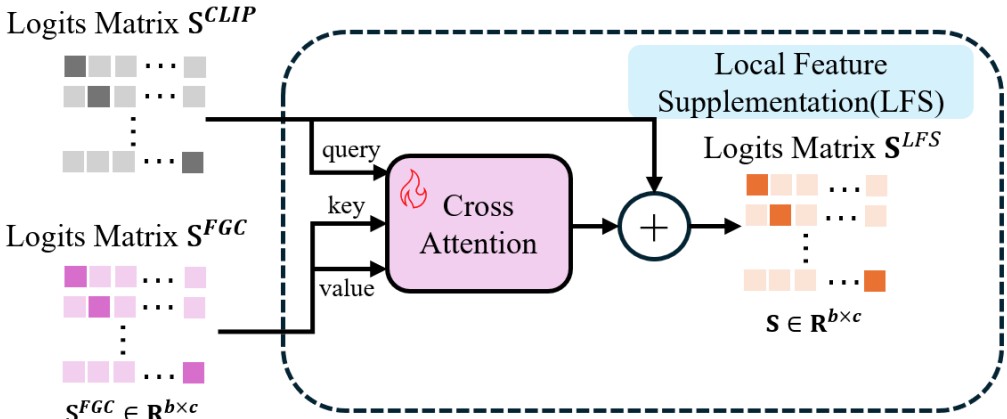

Figure 6: Strategy 3 of LFS

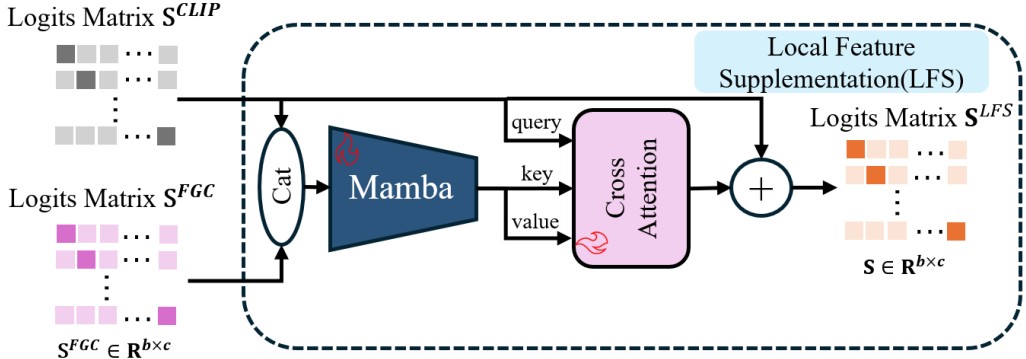

Figure 7: Strategy 4 of LFS

Table 9: Behavior of different strategies on few-shot classification.

| Strategies | ACC | |
| --- | --- | --- |
| | LFS | GF4FC |
| Strategy 1: MLPs | 66.46 | **75.39** |
| Strategy 2: Mamba | 73.25 | 74.55 |
| Strategy 3: CrossAttention | 73.39 | 73.85 |
| Strategy 4: Mamba with CrossAttention | **73.64** | 74.65 |
| Strategy 5: Mamba with AntiAttention | 72.92 | 73.95 |
| Average | 71.93 | 74.48 |

### D.5.5  STRATEGY 5 IN LFS

Shown in figure 8, in Strategy 5, building on Strategy 4, we introduce a novel attention mechanism called AntiAttention. We realized that the cross-attention mechanism in Strategy 4 uses the cosine similarity between query and key to compute weights, which might enhance the existing knowledge in $\mathbf{S}^{CLIP}$ while weakening the missing knowledge. To address this, AntiAttention subtracts the cosine similarity from 1 to compute new weights, enabling the extraction of missing information from $\mathbf{S}^{FGC}$. Strategy 5 simply replaces the CrossAttention in Strategy 4 with AntiAttention.

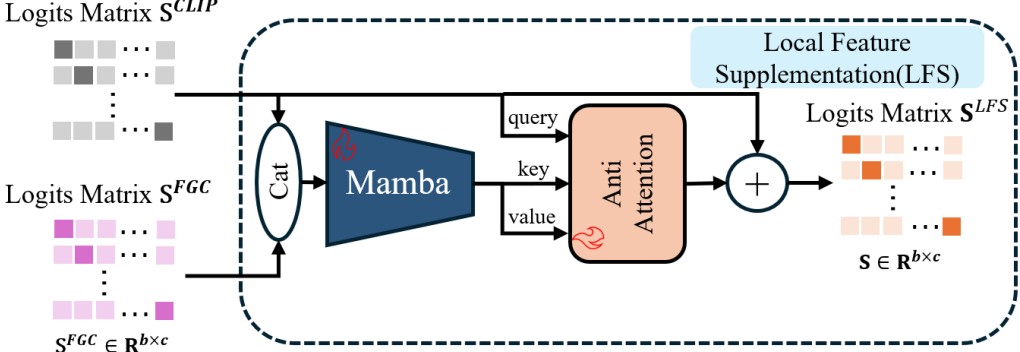

Figure 8: Strategy 5 of LFS

The results indicate that the accuracy of $\mathbf{S}^{LFS}$ is highest at 73.64%, exceeding the average by 1.71%, when using Strategy 4. For $\mathbf{S}^{ALF}$, the accuracy peaks at 75.39% with Strategy 1, surpassing the average by 0.91%. Strategy 1's effectiveness can be attributed to its use of MLPs for knowledge integration. By expanding $\mathbf{S}^{FGC}$ via an MLP, the fine-grained features' structure and usability are enhanced. Projecting $\mathbf{S}^{CLIP}$ to the same dimension using another MLP allows for effective combination of the two feature sets. The third MLP then learns the missing fine-grained features in $\mathbf{S}^{CLIP}$, and the residual connection with $\mathbf{S}^{CLIP}$ supplements local features, making the knowledge generated by Strategy 1 highly effective and usable within the overall architecture. In contrast, other strategies employed more complex structures. Strategy 2 uses a Mamba module to optimize the geometric structure of knowledge in $\mathbf{S}^{FGC}$, while Strategy 3 employs a CrossAttention module to guide the optimization of knowledge manifolds. Strategy 4 combines both the Mamba module and CrossAttention module. However, these complex structures may have led to difficulties in effectively integrating with global information, resulting in lower overall performance. Strategy 5 introduces the AntiAttention mechanism to address the limitations of the CrossAttention mechanism, but it also fails to effectively integrate with global information, leading to suboptimal performance.

Despite Strategy 1 yielding the lowest $\mathbf{S}^{LFS}$ accuracy, it was ultimately chosen due to its superior performance within the overall architecture. This suggests that the knowledge generated by Strategy 1 in $\mathbf{S}^{LFS}$ is the most effective and usable. While other strategies achieved local successes with more complex structures, they failed to effectively integrate with global information.

Our training comprises two phases, each with distinct loss functions to meet different training objectives. For clarity, we first define key parameters and functions. Given a training set with $n$ images across $c$ classes, each image $i \in \{0, 1, \ldots, n-1\}$ has a label $y_i \in \{0, 1, \ldots, c-1\}$. For the image sequence $\mathbf{n}$, the corresponding label vector is $\mathbf{y} = [y_0, y_1, \ldots, y_{n-1}]^T$.

# E  LOSS FUNCTION

## E.1  PROBABILITY DISTRIBUTION

The Sample Probability Distribution Matrix $\mathbf{Y} \in \mathbb{R}^{n \times c}$ is then defined as:

$$\mathbf{Y} = \begin{bmatrix} Y_{0,0} & Y_{0,1} & \cdots & Y_{0,c-1} \\ Y_{1,0} & Y_{1,1} & \cdots & Y_{1,c-1} \\ \vdots & \vdots & \ddots & \vdots \\ Y_{n-1,0} & Y_{n-1,1} & \cdots & Y_{n-1,c-1} \end{bmatrix} \tag{16}$$

where each element $Y_{i,j}$ is:

$$Y_{i,j} = \begin{cases} 1, & \text{if } \mathbf{y}_i = j \\ 0, & \text{otherwise} \end{cases}$$

Output by the model named $N$, $\mathbf{S}^N \in \mathbb{R}^{n \times c}$ is the predicted probability distribution matrix.

## E.2  CROSS-ENTROPY LOSS

Cross-Entropy Loss measures the dissimilarity between predicted and true probability distributions, commonly used for classification tasks. It quantifies the difference between the predicted class probabilities and the true labels, encouraging the model to assign higher probabilities to correct classes. The Cross-Entropy Loss over all $N$ samples and $C$ classes is:

$$\mathcal{L}_{\text{CE}}(\mathbf{S}, \mathbf{Y}) = -\frac{1}{N} \sum_{i=0}^{N-1} \sum_{j=0}^{C-1} \mathbf{Y}_{i,j} \log(\mathbf{S}_{i,j}) \tag{17}$$

## E.3  MEAN SQUARED ERROR (MSE) LOSS

MSE Loss computes the average squared difference between two values. It penalizes larger errors more due to the squaring operation, making it sensitive to outliers. The MSE Loss over all $N$ samples and $C$ classes is:

$$\mathcal{L}_{\text{MSE}}(\mathbf{S}, \mathbf{T}) = \frac{1}{N \times C} \sum_{i=0}^{N-1} \sum_{j=0}^{C-1} (\mathbf{S}_{i,j} - \mathbf{T}_{i,j})^2 \tag{18}$$

## E.4  L1 LOSS

Also known as MAE Loss, L1 Loss computes the average absolute difference between predicted and true values. It is more robust to outliers compared to MSE since it doesn't square the residuals. The L1 Loss over all $N$ samples and $C$ classes is:

$$\mathcal{L}_{\text{L1}}(\mathbf{S}, \mathbf{T}) = \frac{1}{N \times C} \sum_{i=0}^{N-1} \sum_{j=0}^{C-1} |\mathbf{S}_{i,j} - \mathbf{T}_{i,j}| \tag{19}$$

