# OpenReview forum: "Global and Fine-Grained Framework for CLIP with Cross-Modal Mamba in Few-Shot Image Classification"
_ICLR.cc/2026/Conference — ICLR 2026 Conference Withdrawn Submission_

### Official Review · Reviewer_LGEW · 2025-10-20

**Soundness:** 2
**Presentation:** 3
**Contribution:** 2
**Rating:** 2
**Confidence:** 5

**Summary:**

The paper proposes GF4FC, an approach to few-shot image classification that enhances CLIP by integrating dynamic cross-modal interaction and fine-grained visual feature supplementation. It introduces four key modules: (1) CLIMA, which uses a Cross-Modal Mamba (CMM) module to enable bidirectional text-image prompting with linear complexity; (2) FGC, which extracts multi-scale fine-grained features via a VSSM; (3) LFS, which supplements CLIP logits with local features through a residual structure; and (4) ALF, which adaptively fuses global and local logits. Experiments on seven benchmarks show improvements over SOTA CLIP-based FSL methods in higher-shot settings.

**Strengths:**

1) The paper propose GF4FC, which combines Mamba-based State Space Models (SSMs) with cross-modal prompting in CLIP.
2) The paper is clearly structured and the method is easy to conduct.

**Weaknesses:**

1) The motivation of this paper is not clear. In fact, the interaction among modalities is widely explored, such as CLIP-MSA: Incorporating Inter-Modal Dynamics and Common Knowledge to Multimodal Sentiment Analysis With Clip, CMFS: CLIP-Guided Modality Interaction for Mitigating Noise in Multi-Modal
Image Fusion and Segmentation, and CLIP-BCA-Gated: A Dynamic Multimodal Framework for Real-Time Humanitarian Crisis Classification with Bi-Cross-Attention and Adaptive Gating

2) In Table 1, GF4FC performs worse than Tip-Adapter-F in most cases (results are wrongly marked with bold), and the improvement over 16 shots is also limited.

3) Fine-grained and global features are widely used for multimodel alignment. Also, logits-based fusion is a common strategy for multimodal fusion. All these show that the novelty of this paper is limited. The authors should compare these methods and clarify the differences.

4) Some important references are missing, such as Li Y, Xing Y, Lan X, et al. AlignMamba: Enhancing Multimodal Mamba with Local and Global Cross-modal Alignment[C]//Proceedings of the Computer Vision and Pattern Recognition Conference. 2025: 24774-24784.

**Questions:**

1) Could the authors provide qualitative examples (e.g., misclassified images on FGVC-Aircraft) showing how cross-modal interaction harms performance? Is the issue due to misaligned text prompts or Mamba’s sequence modeling bias?
2) Why does the CLIP model perform so badly in Table 2?
3) How about changing the fusion and alignment modules with existing mechanisms (such as CFA [1], XKanFuse[2], CoolNet[3], AlignMamba [4], or [5-6])?

[1] Cross-Modal Fusion and Attention Mechanism for Weakly Supervised Video Anomaly Detection
[2] XKanFuse: A novel cross-modal fusion method based on Kolmogorov-Arnold Network for multi-modal medical image fusion
[3] Cross-modal fine-grained alignment and fusion network for multimodal aspect-based sentiment analysis
[4] AlignMamba: Enhancing Multimodal Mamba with Local and Global Cross-modal Alignment
[5] Wan Y, Wang W, Zou G, et al. Cross-modal feature alignment and fusion for composed image retrieval[C]//Proceedings of the IEEE/CVF Conference on Computer Vision and Pattern Recognition. 2024: 8384-8388.
[6] Mingyong L, Yewen L, Mingyuan G, et al. CLIP-based fusion-modal reconstructing hashing for large-scale unsupervised cross-modal retrieval[J]. International Journal of Multimedia Information Retrieval, 2023, 12(1): 2.

---

### Official Review · Reviewer_WHU7 · 2025-10-31

**Soundness:** 2
**Presentation:** 2
**Contribution:** 2
**Rating:** 2
**Confidence:** 4

**Summary:**

This paper proposes a framework that enhances CLIP's performance on FSL by performing interaction between image and text modalities and fusing global and local visual features to capture fine-grained information.

**Strengths:**

1. The paper is clearly written and easy to understand.

2. The framework comprehensively considers text–image modality interactions and integrates global and local information effectively.

**Weaknesses:**

1. The main contribution appears to be a combination of existing methods, including Mamba-based attention and global--local feature fusion.
2. The experiments mainly compare against works before 2023, which is insufficient

3. If I understand correctly, the performance of the proposed method is worse than Tip-Adapter-F in most settings (1--8 shots), and only slightly better in the 16-shot setting. I think such performance is unacceptable for few-shot scenarios.

| Methods       | 0-Shot | 1-Shot | 2-Shot | 4-Shot | 8-Shot | 16-Shot |
|----------------|:------:|:------:|:------:|:------:|:------:|:-------:|

| Tip-Adapter-F|   -    | 71.99  | 73.52  | 76.71  | 79.57  | 81.86   |

| GF4FC |               -    | 71.26 | 72.09 | 76.27 | 79.55 | 81.95 |

**Questions:**

Please see the weaknesses.

---

### Official Review · Reviewer_vPEZ · 2025-11-01

**Soundness:** 1
**Presentation:** 1
**Contribution:** 1
**Rating:** 2
**Confidence:** 3

**Summary:**

This paper introduces GF4FC, a method that integrates Mamba and several specially designed modules to enhance the final performance in few-shot image classification. The whole story lacks of strong motivation and the model have small impacts.

**Strengths:**

- Experiments are conducted on 7 datasets, and ablation studies are made.
- The proposed method surpasses other baselines, but they are not the latest ones.

**Weaknesses:**

- The research problems (or the issues of related works) lack of in-depth analyses, and all the issues (classification boundaries cannot be optimized, underutilizing cross-modal interactions, neglecting dynamic interaction) are superficial.
- Due to the superficial summarization of existing problems, the motivations are significantly weakened. The introduction of Mamba, VSSM, and corresponding modules do not make sense. It seems to be that the authors tried so and found these modules worked, so they made up a story.
- The baselines are ancient (2021, 2022, and 2023) and I cannot tell if the final results are really SOTA performance.
- This paper lacks of significant analysis.
    - If granularity is a big contribution of your method, please design metrics to demonstrate its effectiveness.
    - If Mamba really works on this, please explain why it works by specially designed experiments or analysis rather than just showing the final performance gaps.

**Questions:**

- Vssm lacks of proper definition when it first appears.

---

### Official Review · Reviewer_TEZR · 2025-11-06

**Soundness:** 2
**Presentation:** 2
**Contribution:** 2
**Rating:** 2
**Confidence:** 4

**Summary:**

See Questions

**Strengths:**

See Questions

**Weaknesses:**

See Questions

**Questions:**

After reading the paper, I have the following comments and suggestions that I hope the authors could take seriously:

- Q1. Writing issues. The writing quality of the paper requires significant improvement. For example:

-- (a) In the abstract, Cross-modal Mamba is introduced abruptly without any prior context or motivation. It's unclear why Mamba is introduced at all, and its relevance is not properly established.

-- (b) In the second paragraph of the introduction, the authors categorize few-shot learning methods into three types. However, in the second type, the focus shifts to adapters for preventing catastrophic forgetting, which deviates from the main topic of few-shot learning. The explanation should be aligned with the few-shot learning context. Moreover, the discussion seems to imply that adapters are only applicable to CLIP, which is inaccurate. Also, the example of “a dog eating” appears to be irrelevant to the paper's core content and adds to the confusion.

- Q2. Lack of clear motivation for using Mamba. The motivation for introducing Mamba in this work is unclear. It appears that Mamba is used merely to "jump on the bandwagon", rather than being a well-justified design choice. The paper claims to adopt an SSM-based approach but does not provide sufficient explanation or reasoning for this decision.

- Q3. Limited novelty. The proposed contributions, such as integrating global feature enhancement with local feature supplementation, or leveraging SSM-based methods like Mamba and SSM, are not particularly novel. Similar ideas have already been extensively explored in existing literature. The contributions of this paper do not stand out as innovative.

- Q4. Insufficient related work review. CLIP was proposed in 2021, and since then, a large body of research has focused on few-shot learning based on CLIP. While this paper claims to propose a CLIP-based method, it lacks a comprehensive review of relevant prior work. It only briefly mentions some basic approaches such as CoOp and CoCoOp. Additionally, the reference list contains only around 28 citations, which is far too limited for a paper in the few-shot learning domain. This suggests the authors are a lack of deep understanding of the field.

- Q5. Overuse of acronyms and jargon. The paper introduces numerous method names and acronyms (e.g., GF4FC, CLIMA, FGC, LFS, CMM, ALF), which makes the reading experience overwhelming and confusing. Many of these terms do not appear to be essential or meaningful.

- Q6. Other issues.

-- (a) The punctuation usage is inconsistent. For instance, proper quotation marks (“ ”) should be used instead of incorrect ones (" ").

-- (b) The mathematical expressions are poorly formatted and appear unprofessional, indicating inexperience in technical writing.

---

### Note · Authors · 2025-11-13

I have read and agree with the venue's withdrawal policy on behalf of myself and my co-authors.